# Pfizer-BioNTech mRNA Vaccine Protection among Children and Adolescents Aged 12–17 Years against COVID-19 Infection in Qatar

**DOI:** 10.3390/vaccines11101522

**Published:** 2023-09-25

**Authors:** Khadieja Osman, Jesha Mundodan, Juel Chowdhury, Rejoice Ravi, Rekayahouda Baaboura, Jeevan Albuquerque, Bilal Riaz, Reem Yusuf Emran, Khatija Batoul, Abdul Mahmood Esameldin, Zinah Al Tabatabaee, Hayat Khogali, Soha Albayat

**Affiliations:** 1National COVID-19 Track & Trace Team, Ministry of Public Health, Doha P.O. Box 42, Qatar; dr.khadijahassan@gmail.com (K.O.); juel.chowdhry@moph.gov.qa (J.C.); rravi@moph.gov.qa (R.R.); rekayahouda.baaboura@moph.gov.qa (R.B.); jalbuquerque@moph.gov.qa (J.A.); briaz@moph.gov.qa (B.R.); remran@moph.gov.qa (R.Y.E.); hkhogali@moph.gov.qa (H.K.); salbayat@moph.gov.qa (S.A.); 2Vaccination Section, Ministry of Public Health, Doha P.O. Box 42, Qatar

**Keywords:** adolescent, children, 12–17 years COVID-19 vaccine, fully vaccinated, partially vaccinated, vaccination status, RT-PCR-positive, Pfizer, BNT162b2, vaccine effectiveness

## Abstract

Qatar was also hit hard by the global pandemic of SARS-CoV-2, with the original virus, Alpha variant, Beta variant, Omicron BA.1 and BA.2 variants, Omicron BA.4 and BA.5 variants, and Delta variant, sequentially. The two-dose primary series of BNT162b2 (Pfizer-BioNTech) COVID-19 vaccine against SARS-CoV-2 infection has been approved for use in 30 µg formulations among children and adolescents aged 12–17 years as of 16 May 2021. This study aimed at estimating the effectiveness of the 30 µg BNT162b2 Pfizer-BioNTech mRNA COVID-19 vaccine against the pre-Omicron variants of SARS-CoV-2 infection in children and adolescents aged 12–17 years residing in Qatar. A test-negative matched case-control study was conducted. The subjects included any child or adolescent aged 12–17 years who had been tested for SARS-CoV-2 using RT-PCR tests performed on nasopharyngeal or oropharyngeal swabs, as part of contact tracing, between June and November 2021, and was eligible to receive the BNT162b2 vaccine as per the national guidelines. Data regarding 14,161 children/adolescents meeting inclusion–exclusion criteria were retrieved from the national Surveillance and Vaccine Electronic System (SAVES). Of the total, 3.1% (444) were positive for SARS-CoV-2. More than half (55.96%) were vaccinated with two doses of Pfizer-BioNTech-mRNA COVID-19 vaccine. Amongst those immunized with two doses, 1.2% tested positive for SARS-CoV-2, while 5.6% amongst the unvaccinated tested positive. The vaccine effectiveness was calculated to be 79%. Pfizer-BioNTech mRNA COVID-19 vaccine provides protection from COVID-19 infection for children/adolescents; hence, it is crucial to ensure they receive the recommended vaccines.

## 1. Introduction

The COVID-19 pandemic was an unprecedented health emergency and could be controlled to an extent by a stable public health system through early containment by early detection of infected persons, the isolation of infected cases, as well as contact tracing, testing, and quarantine of these contacts [1]. In addition, nonpharmaceutical preventive health measures, such as hand washing, using face masks, physical distancing, stay-at-home orders, school and venue closures, workplace restrictions, and environmental cleaning, were adopted internationally. These responses were modified, changed, or intensified with the emergence of new epidemiologic data, experience-sharing from other countries, and emerging newer variants.

Qatar experienced five waves of SARS-CoV-2 infection, by the index virus [2], the B.1.1.7 (Alpha) variant [3], the B.1.351 (Beta) variant [4], the B.1.1.529 (Omicron) subvariants BA.1 and BA.2 [5], the Omicron subvariants BA.4 and BA.5 [6], and a prolonged phase of the B.1.617.2 (Delta) variant [7], sequentially. Community transmission of the SARS-CoV-2 Delta variant (B.1.617.2) was first identified in Qatar by the end of March 2021 [2,8,9]. Although Delta variant incidence increased and reached about 200 cases per day in the summer of 2021, it remained low compared to earlier variants incidences. Between 23 March 2021 and 7 September 2021, 43.0% of diagnosed infections were Delta variant infections [2,4]. The first Omicron variant infection in Qatar was identified on 24 November 2021. Within four weeks, it became the predominant strain [10].

Free SARS-CoV-2 testing was widely available in Qatar and was required for those in close contact with an infected person, with symptoms such as fever or acute respiratory illness, as well as people returning from travel abroad. All specimens collected via nasopharyngeal/oropharyngeal swab, irrespective of where they were collected, be it private or public health facilities, were tested at the National Virology laboratory, using real-time PCR tests, following the national testing standards.

However, vaccination remains the most efficient and effective control strategy against COVID-19. Global vaccine development efforts have been accelerated in response to the devastating COVID-19 pandemic, like accelerated evaluation procedures and authorization for emergency use. Several pharmaceutical companies were trying to develop an effective vaccine against severe acute respiratory syndrome coronavirus 2 (SARS-CoV-2) infection. Phase III trials reported high vaccine effectiveness (VE) against SARS-CoV-2 infection, with 70.4% for Oxford-AstraZeneca vaccine (ChAdOx1 nCoV-19 vaccine) [4], 95.0% effectiveness with Pfizer BioNTech vaccine (BNT162b2 mRNA vaccine) [5], and 94.1% with Moderna vaccine (mRNA-1273 vaccine) [6].

Qatar was among the first Gulf Cooperation Council (GCC) countries to procure COVID-19 vaccines and start the COVID-19 vaccination campaign nationwide for the citizens and residents. The COVID-19 vaccination campaign plan was developed by the National Strategic Committee and implemented by the Health Protection and Communicable Diseases Department (HP-CDC) of the Ministry of Public Health (MOPH) along with Hamad Medical Corporation (HMC) and Primary Health Care Corporation (PHCC). Four COVID-19 vaccines, namely, Pfizer, Moderna, AstraZeneca, and Jansen & Jansen, were approved in Qatar. A few others, like Sinopharm, Sputnik, and Sinovac, were conditionally approved. Guidelines and recommendation for COVID vaccines were prepared and distributed and are also regularly updated with the development of new scientific evidence globally.

In Qatar, vaccination commenced on 23 December 2020, primarily with the Pfizer BNT162b2 mRNA vaccine. The Moderna mRNA-1273 vaccine was introduced later. Vaccines were provided free of cost to all nationals and residents of Qatar through the public health care system and mass vaccination centers such as Qatar National Convention Centre (QNCC), VCIA (Vaccination Center Industrial Area), and Qatar Vaccination Centre (QVC). The primary focus or target groups were the high-risk groups, namely, frontline health care workers, those with chronic illnesses, and the elderly population. Once almost half the primary targets were covered, other categories like teachers and the workers living in close proximities and dormitories were focused through VCIA and QNCC. The mass vaccination campaigns for the general public helped to effectively increase the vaccine coverage for COVID-19 in Qatar, which helped reduce the Delta variant transmission in the community.

The highlight was that Qatar provided free vaccination for all. All entities providing vaccine were mandated to enter the vaccination details in the national vaccine registry (SAVES). A well-established Adverse Event Following Immunization (AEFI) reporting platform existed for the clinicians to report any suspected AEFI. In addition to this reporting platform for physicians, a link was provided on the ministry’s website so that any individual experiencing any adverse event following COVID vaccination could register, and these data was analyzed monthly. Hence, it was easy to track adverse events following COVID vaccine administration. Only eight AEFIs were reported among those 12–17 years old post-vaccination with Pfizer BioNTech vaccine, of which five were related to the first dose. Most of the reported adverse events were mild reactions. Two of them were severe reactions—one was a case of myocarditis and the other anaphylactic reaction—both following the second dose, and these cases were hospitalized. However, there no deaths reported following vaccination. Hence, the Pfizer BioNTech vaccine in this age group was deemed safe, which, in turn, increased the uptake.

More than 80% of the resident population in Qatar had completed the primary series with either BNT162b2 (Pfizer-BioNTech) vaccine or the mRNA-1273 (Moderna) vaccine by September 2021 [5,6,7]. Pfizer-BioNTech (BNT162b2) and Moderna (mRNA-1273) mRNA-based vaccines are given as two doses scheduled three to four weeks apart, keeping a minimum of 15 days between the two doses.

As the vaccination was scaled up, the country faced two back-to-back waves of SARS-CoV-2 from January 2021 to June 2021, which predominantly were B.1.1.7 (Alpha) and B.1.351 (Beta) variants [6,9,11]. Community transmission of the B.1.617.2 (Delta) variant was first detected towards the end of March 2021, and it had become the dominant strain circulating by the summer [12,13,14].

While children tend to experience less symptomatic SARS-CoV-2 infection than adults, it is important to note that schools, youth sports, and other community events can still contribute to outbreaks and transmission. These settings can pose a significant risk even with high adult immunization rates [15]. The absence of in-person learning during the pandemic has had a detrimental impact on children. Given the vaccine’s favorable safety and side-effect profile, high efficacy, and acceptable risk-to-benefit ratio in adolescents, evaluating its effectiveness in younger age groups is justified. Vaccinating adolescents can enable them to safely return to in-person learning and reintegrate into society, addressing the debilitating mental health consequences of the COVID-19 pandemic [16,17]. Reducing COVID-19-related morbidity and mortality in adolescents can be achieved by administering a safe and effective vaccine. Additionally, the availability of effective vaccines for adolescents is crucial in decreasing the reservoir of SARS-CoV-2. In line with this, the BNT162b2 vaccine has received emergency use authorization for adolescents aged 12 to 15 [18].

The World Health Organization’s (WHO) Strategic Advisory Group of Experts (SAGE) on immunization updated the roadmap for prioritizing COVID-19 vaccines on 21 January 2021. Children and adolescents with comorbidities were identified as medium-priority population groups for administering the primary series and booster doses. In contrast, healthy children and adolescents were identified as the low-priority use group because of their low risk of severe disease, hospitalization, and fatality. European Union countries recommend primary vaccination against COVID-19 for 12–17-year-olds [7].

Ministry of Public Health (MOPH) Qatar approved Pfizer-BioNTech (BNT162b2) COVID-19 vaccine administration to children and adolescents, based on regional and global studies showing its safety and efficacy in this age group [8,9]. The two-dose primary series of BNT162b2 (Pfizer-BioNTech) COVID-19 vaccine against SARS-CoV-2 infection has been approved for use in 30 µg formulations among children and adolescents aged 12–17 years as of 16 May 2021 and 10 µg formulations among children aged 5–11 years as of 30 January 2022.

The objective of this study was to assess the effectiveness of the 30 μg dose of BNT162b2 COVID-19 vaccine (Pfizer) against severe acute respiratory syndrome coronavirus 2 (SARS-CoV-2) infection among children and adolescents aged 12–17 years, in Qatar, before the emergence of the Omicron variant.

## 2. Materials and Methods

*Study design*: a test-negative matched case-control study design [11,19,20].

*Study population*: children and adolescents aged 12–17 years, residing in Qatar, who had undergone COVID-19 tests using reverse transcription polymerase chain reaction (RT-PCR) for SARS-CoV-2 on nasopharyngeal swabs (NPS) or oropharyngeal swabs (OPS) as part of contact tracing, between 1 June and 30 November 2021. This ensured that the first batch of children/adolescents would have received both doses of the primary series and excluded Omicron variant-positive cases from the analysis. RT-PCR testing detects SARS-CoV-2 RNA at low levels, with analytic sensitivity of (98%) and specificity of (97%) [21].

*Cases:* children/adolescents aged 12–17 years with positive test results on RT-PCR for SARS-CoV-2.

*Controls*: children/adolescents aged 12–17 years who had negative test results on RT-PCR for SARS-CoV-2, matched by calendar week for the RT-PCR tests.

### 2.1. Inclusion Criteria


Any children and adolescents aged 12–17 years residing in Qatar tested for SARS-CoV-2 using RT-PCR between 1 June 2021 and 30 November 2021 were eligible irrespective of nationality, gender, and vaccination status.Children and adolescents eligible to receive the Pfizer-BioNTech mRNA COVID-19 as per Ministry of Public Health (MOPH) guidelines.


### 2.2. Exclusion Criteria


Children and adolescents tested for COVID-19 using a method other than RT-PCR.Uncertainty about the COVID-19 test results, which includes “Inconclusive” results or if results were unavailable for any reason.


*Sampling*: A total of 14,298 children and adolescents aged 12–17 years who were tested for COVID-19 in the date range of the study, vaccinated or unvaccinated, were extracted from the national database. Of this total, 137 had inconclusive test results and, hence, were removed, leaving us with 14,161 children and adolescents between 12 to 17 years, irrespective of vaccination status, who were included in the study. The study population selection process is illustrated in Figure 1.

The effect modification of the vaccine effectiveness by differences in the variants exposed to, changes in testing frequency over time, and differences in infection-derived immunity among the unvaccinated were adjusted by matching the cases and controls by calendar week for the RT-PCR test [22,23,24,25]. It was possible to find PCR-negative matches for most age groups due to the higher number of PCR-negative test results than PCR-positive results. Nonpharmaceutical interventions (NPIs), including face masks, social distancing, hand washing, and hand hygiene, were mandated in Qatar during the study period, with varying levels of restrictions for the public as per MOPH guidelines; hence excluding the confounding effect expected due to change in behavior after vaccination. Cases and controls were matched in a 1:5 ratio to maximize statistical power.

*Data collection*: COVID-19 case investigation teams receive a list of laboratory-confirmed SARS-CoV-2 cases from various government and private sectors nationwide. Team members contact the index case by phone to obtain the necessary details and record the number and details of those the patient had close contact with in the last 48 h. A close contact refers to anyone who has met someone infected with the COVID-19 virus, starting from 2 days before the onset of the infected person’s illness up to 14 days after. Based on this information, a swabbing dispatch list was prepared daily, which is then forwarded to the field team supervisor for action. The swab team successfully visited the homes and workplaces of these confirmed cases and collected the necessary swabs from the close contacts enlisted. It was recommended to collect nasopharyngeal and oropharyngeal swabs in a single vial containing transport medium and only oropharyngeal swabs for children under 14 years of age.

Nasopharyngeal and oropharyngeal swabs collected across the country (irrespective of public/private), including those by the field teams, were tested using the RT-PCR tests for SARS-CoV-2 at the National Virology Laboratory under Hamad Medical Cooperation (HMC). The MOPH database, Surveillance and Vaccine Electronic System (SAVES), receives all real-time RT-PCR test results from the laboratory [10,26]. The data regarding COVID-19 laboratory testing, vaccination (which includes the types of vaccine and dates of the first and second doses of vaccine administration, place of immunization, expiry date of the vaccine, and the lot number), and associated demographic information were retrieved from the national integrated digital health information platform, Surveillance and Vaccine Electronic System (SAVES), owned by the Ministry of Public Health (MOPH), Qatar. The vaccination details of the citizens, residents, and visitors who had been vaccinated abroad were incorporated into the National Vaccine registry upon arrival in Qatar [27].

The study participants were divided into 3 categories based on vaccination status: fully vaccinated and immune (those who had completed 14 days after receiving the second dose of vaccine), partially vaccinated or partially immune (participants who had received only 1 dose of vaccine or those who had not completed 14 days after receiving the second dose of vaccine) and unvaccinated (participants who had not received any dose of the vaccine).

*Data analysis*: a total of 14,161 children and adolescents aged 12–17 years who were tested for COVID-19 in the date range of the study, vaccinated or unvaccinated, were included in the study. The case and control groups were described using frequency distribution. The vaccine effectiveness of the BNT162b2 Pfizer vaccine among children/adolescents aged 12–17 years at least 14 days post-second dose was estimated by calculating relative risk reduction (RRR). The differences in VE of the COVID-19 vaccine according to age, gender, nationality (Qatari and non-Qatari), and vaccination status (fully vaccinated, partially vaccinated, unvaccinated) were also analyzed. VE based on number of days from receipt of second dose was also analyzed, because people who were vaccinated earlier are at increased risk for infection compared to those vaccinated later.

## 3. Results

Table 1 shows the characterization of the 14,161 children and adolescents aged 12–17 years included in the study by age, gender, nationalities, and vaccination status. The majority (40.6%) were 12–13 years old. The male to female proportions were nearly the same. A majority (60.9%) of the study participants were non-Qataris, as expected from the population distribution of Qatar. A total of 7925 (55.96%) were vaccinated with two doses of the BNT162b2 vaccine, and 6225 (43.96%) were unvaccinated. This higher proportion of vaccinated participants can be explained by the fact that as of 31 May 2021, more than half of residents had received at least one dose and 41% had completed both doses.

A majority (65.8%) of the younger age group (12–13 years) were unvaccinated, while three-fourths (75.1%) of those aged 16–17 were fully vaccinated. Hence, Table 1 shows a significant (*p*-value < 0.001) association between age and completion of the primary series of vaccines; that is, as the age increases, the proportion of fully vaccinated children increases. (Table 1) This can be explained by the fact the older age group were enthusiastic to get vaccinated as this would give them the privilege to go out into malls and restaurants. There was no significant difference between the genders with regards to the vaccination status.

No significant difference in vaccination status was noted between the nationals and non-nationals. A similar proportion of nationals and non-nationals were vaccinated with at least one dose. This may be explained by the fact that the government provided COVID vaccines free of cost universally to both nationals and non-nationals.

According to the data presented in Table 2, only 3.1% of the study population were infected and, among the infected, 44.14% were aged 12–13, whereas only half this proportion (20.7%) were of the older age group (16–17 years). The likelihood of testing positive for COVID-19 decreases with increasing age and vaccination status. This is in sync with the vaccination rates among the different age groups.

There was a significant (<0.001) difference between the cases and controls with regards to nationality. More than half (53.4%) of the cases were non-Qataris, in comparison to 46.6% Qataris. Similarly, a higher proportion of the control group were non-Qataris (61.2%). This can be explained by the population distribution of the residents of Qatar.

However, no significant difference was found between the cases and controls with regards to gender. The males to female proportion was similar in both case and control groups.

Table 3 shows that out of 7925 fully vaccinated subjects, only 97 (1.2%) tested positive for SARS-CoV-2 by RT-PCR, whereas 346 (5.6%) tested positive among the 6225 in the unvaccinated population. Among the case group, 21.8% were fully vaccinated, while 77.9% were unvaccinated. Similarly, among the control group, 57.2% had received at least one dose of COVID vaccine, while 42.8% were unvaccinated. There is a significant difference in the proportion of fully vaccinated between the cases and controls (*p* < 0.001).

Relative risk reduction (RRR) was calculated as:$$\frac{\mathrm{vaccinated}\;\mathrm{among}\;\mathrm{cases}\;\times \;\mathrm{unvaccinated}\;\mathrm{among}\;\mathrm{controls}}{\mathrm{vaccinated}\;\mathrm{among}\;\mathrm{controls}\;\times \;\mathrm{unvaccinated}\;\mathrm{among}\;\mathrm{cases}}=0.21$$Vaccine effectiveness (VE) = 1 − RRR = 79.0%

Figure 2 shows that, out of a total of 98 individuals who received the vaccine and tested SARS-CoV-2 positive, 2 (1.9%) tested positive within 14 days of the first dose, while 3 (2.8%) tested positive between 15 and 30 days after the first dose. Another 2.8% tested positive within 14 days of receiving the second dose. It is seen that nearly half of the fully vaccinated participants (48.1%) tested positive after 90 days following the second dose, while only 15.7% tested positive within 31–60 days of receiving the second dose, and 17.6% were infected between 61–90 days.

## 4. Discussion

The effectiveness against Alpha and Beta variants was high: over 75% in Qatar [8,28,29,30]. On the other hand, the effectiveness against Delta variant infection seven or more days after the second dose was (55.5%) (95% CI, 51.2–59.4%), irrespective of the vaccine type, 51.9% (95% CI, 47.0–56.4%) with BNT162b2, and 73.1% (95% CI, 67.5–77.8%) with mRNA-1273, specifically. The protection was higher 14 days after the second dose of the primary series. Delta variant’s effectiveness was evaluated several months after the second dose for the residents [31].

The present study estimated the effectiveness of BNT162b2 (Pfizer) vaccine against SARS-CoV-2 infection, in children/adolescents aged 12–17 years, 14 or more days after the second dose, to be 79% in Qatar during the pre-Omicron period. Protection offered by the vaccine against SARS-CoV-2 infection among vaccinated children was higher among the older age group (*p*-value < 0.001). The VE showed a gradual decline in immunity over time following the second dose.

Another study carried out in Qatar during the pre-Omicron period found that the vaccine effectiveness against SARS-CoV-2 infection among adolescents was 87.6% (95% CI, 84.0 to 90.4). The level of protection was approximately 95% post-second dose and declined slowly over time but remained above 50% for at least five months [32].

The findings are consistent with evidence from other countries regarding protection provided by the vaccine in preventing SARS-CoV-2 infection among children and adolescents [33,34,35,36,37,38]. The vaccine demonstrated efficacy similar to that observed in young adults [39].

Testing for SARS-CoV-2 is performed on a mass scale in Qatar [40,41]. About three-fourths of cases are diagnosed because of routine screening tests and not because of the appearance of symptoms [42]. Since the hospitalization and deaths were low among the younger population, it was difficult to differentiate whether the protection was offered by the natural infection or by vaccination with the mRNA vaccines [40,41,42,43,44].

In a study carried out in Italy, the fully vaccinated group had a vaccine effectiveness of 29.4% (95% CI 28.5–30.2) and 41.1% (95% CI 22.2–55.4) with BNT162b2 (Pfizer-BioNTech) against SARS-CoV-2 infection and severe SARS-CoV-2 infection, respectively. The partially vaccinated group had a vaccine effectiveness of 27.4% (95% CI 26.4–28.4) against SARS-CoV-2 infection and 38.1% (95% CI 20.9–51.5) against severe SARS-CoV-2 infection. The vaccine effectiveness was highest, at 38.7% (95% CI 37.7–39.7%), within the first 14 days after completing the primary series. However, it decreased to 21.2% (95% CI 19.7–22.7%) between 43 and 84 days after being fully vaccinated with two doses [33].

In a retrospective cohort study, from Singapore, the estimated vaccine effectiveness (VE) against all COVID-19 infections in the age group of 12–18 years following two doses of BNT162b2 (Pfizer-BioNTech, New York, NY, USA) vaccines was 59% (95% CI: 55–63%) over the period of Delta variant dominance from 1 June to 20 November 2021 [39]. In a US study, during the Omicron-dominant period, the vaccine effectiveness was 59.0% (95% CI 22.0–79.0) 14–149 days after receipt of the second dose among adolescents aged 12–15 years [45].

The VISION Network study carried out across 10 states of the United States during 26 August 2021–22 January 2022 found that vaccine effectiveness after receipt of both two and three doses was lower during the Omicron-predominant period when compared to the Delta-predominant period. During both periods, VE waned with increasing time since vaccination. During the Omicron period, VE during the first two months after a third dose was 87% against emergency department/OPD clinic visits, and the VE decreased to 66% among those vaccinated who were vaccinated four to five months earlier. VE against hospitalizations was 91% during the first two months following a third dose and decreased to 78% beyond four months after a third dose [46].

A study conducted in England showed that vaccine effectiveness was 76.3% (95% CI 61.1–85.6%) 28 days after the first dose for those aged 16–17 years and 83.4% (54.0–94.0%) for those aged 12–15 years. The first dose of the vaccine was most effective for 16–17-year-olds against symptomatic disease caused by the Delta variant between days 14 and 20, with a peak effectiveness of 75.9% (95% CI: 74.3–77.3). However, effectiveness gradually decreased to 29.3% (25.9–32.6) between days 84 and 104. Among children/adolescents aged 12–15 years and 16–17 years, the VE against Delta infection showed a peak of 68% (95% CI: 64–71%) and 62% (95% CI: 57–66%), respectively, on days 21–48 after the first dose. Among those aged 16–17 years who received both doses, the VE against infection with the Delta variant was highest, at 93% (95% CI: 90–95%), between days 35 and 62 after vaccination but declined to 84% (95% CI: 76–89%) after 63 days [47].

Out of the 991,682 children and adolescents in Denmark who underwent RT-PCR testing for SARS-CoV-2, 7.5% (74,611) tested positive. Compared to unvaccinated adolescents, those who received one dose of the vaccine had an estimated effectiveness of 62% (with a 95% confidence interval of 59% to 65%) after 20 days. After 60 days, the estimated effectiveness of two doses was 93% (with a 92% to 94% confidence interval) during a period when the Delta variant was the most prevalent. The BNT162b2 vaccine demonstrated high effectiveness, of 93%, against SARS-CoV-2 infection among adolescents aged 12–17 years 60 days after receiving the second dose [48].

## 5. Strengths

Testing for SARS-CoV-2 was carried out on a mass scale in Qatar, and the results are tracked centrally. Nearly all testing in the country was via RT-PCR during this period. Universal access to the vaccines was provided to the eligible population free of cost. The National Vaccine registry facilitated obtaining the vaccination details of any individual vaccinated in the country. Additionally, since most of the testing was routine, contacts were likely found to be asymptomatic. These facts would suggest that the vaccine was efficacious against infection and not symptomatic infection/hospitalization/those seeking health care.

## 6. Limitations

Nearly all testing in the country was via RT-PCR during this period; however, there may have been a minority group who would have gone for home rapid testing kits. Since our study design relies on people being tested via RT-PCR only, these home tests were not included; moreover, a home test did not result in the same contract tracing as the PCR; hence, we may be underestimating cases. However, since nearly all testing in the country is performed using RT-PCR, these will not constitute huge numbers. Confounders like ethnicity and the presence of comorbidity have not been taken into consideration in this study. This study does not include the effectiveness of additional doses of the COVID-19 vaccine in severely immunocompromised children and adolescents, for whom additional doses should be considered as part of the primary vaccination schedule.

## 7. Conclusions

The BNT162b2 vaccine was associated with high protection against SARS-CoV-2 disease in children and adolescents. At this stage, priority should be given to completing the primary vaccination course for all the eligible population. Taking into consideration the waning of immunity, attention should be given to providing additional doses to high-risk and priority groups, according to national recommendations.

Unvaccinated persons are more likely to be infected when compared to vaccinated individuals, leading to higher incidence among the unvaccinated. Thus, Qatar began with easing restrictions for those vaccinated; however, now there is concern about a potentially increased risk of exposure among vaccinated individuals. Due to their perceived lower risk, the vaccinated may have adhered less strictly to safety measures such as face masks.

## Figures and Tables

**Figure 1 vaccines-11-01522-f001:**
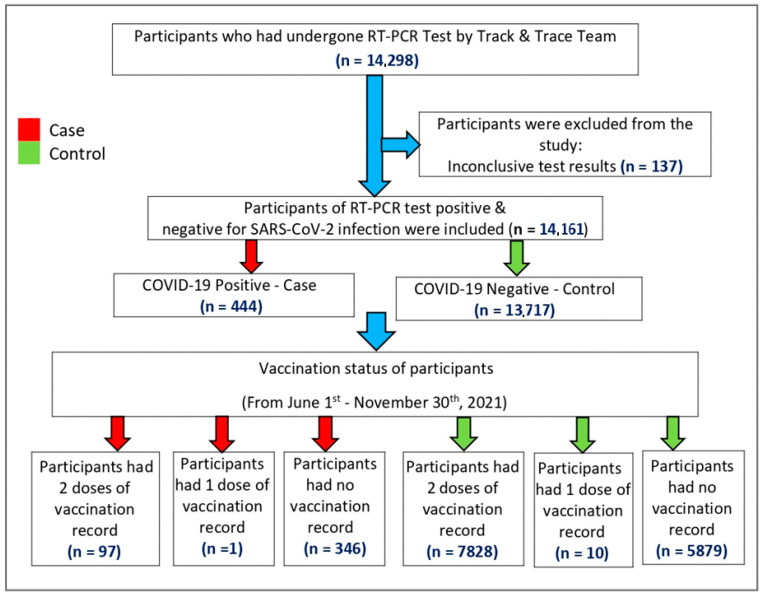
Flowchart illustrating the selection process of participants for investigating Pfizer-BioNTech mRNA vaccine effectiveness against SARS-CoV-2 infection.

**Figure 2 vaccines-11-01522-f002:**
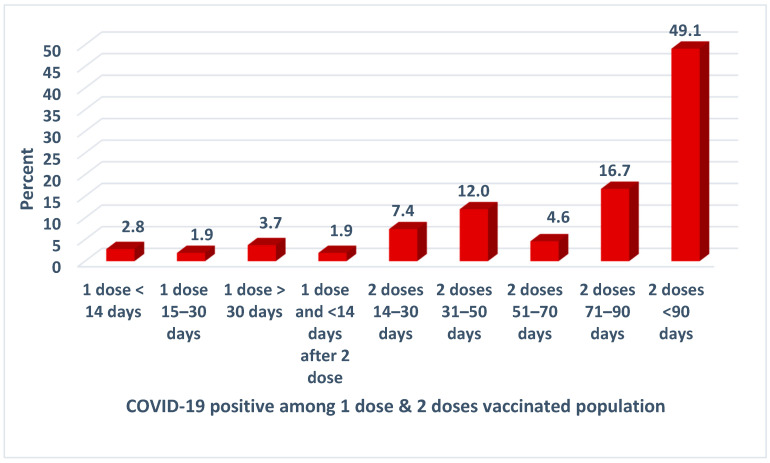
COVID-19 breakthrough infections among vaccinated study participants.

**Table 1 vaccines-11-01522-t001:** Description of the study participants by age, gender, nationalities, and vaccine status.

	Fully Vaccinated (Two Doses)	Partially Vaccinated (One Dose)	Unvaccinated	Total	*p*-Value
Distribution by Age group (in years)					
12–13	1962 (34.1%)	2 (0.03%)	3787 (65.8%)	5751 (40.6%)	<0.001
14–15	3222 (67.7%)	6 (0.13%)	1534 (32.2%)	4762 (33.6%)	
16–17	2741 (75.1%)	3 (0.08%)	904 (24.8%)	3648 (25.8%)	
Total	7925 (55.96%)	11 (0.08%)	6225 (43.96%)	14,161 (100.0%)	
Distribution by Gender					
Male	3840 (54.1%)	4 (0.06%)	3253 (45.8%)	7097 (50.1%)	2.753
Female	4085 (57.8%)	7 (0.1%)	2972 (42.1%)	7064 (49.9%)	
Total	7925 (55.96%)	11 (0.08%)	6225 (43.96%)	14,161 (100.0%)	
Distribution by Nationalities					
Qataris	2914 (52.7%)	3 (0.05%)	2616 (47.3%)	5533 (39.1%)	1.202
Non-Qataris	5011 (58.1%)	8 (0.09%)	3609 (41.8%)	8628 (60.9%)	
Total	7925 (55.96%)	11 (0.08%)	6225 (43.96%)	14,161 (100.0%)	

**Table 2 vaccines-11-01522-t002:** Distribution of cases and controls by age, gender, and nationalities.

	COVID-19-Positive (Cases)	COVID-19-Negative (Control)	Total	*p*-Value
Distribution by Age group (in Years)				
12–13	196 (44.1%)	5555 (40.5%)	5751	0.044
14–15	156 (35.1%)	4606 (33.6%)	4762	
16–17	92 (20.8%)	3556 (25.9%)	3648	
Total	444 (100%)	13,717 (100%)	14,161	
Distribution by Gender				
Male	224 (50.5%)	6873 (50.1%)	7097	0.886
Female	220 (49.5%)	6844 (49.9%)	7064	
Total	444 (3.1%)	13,717 (100%)	14,161	
Distribution by Nationalities				
Qataris	207 (46.6%)	5326 (38.8%)	5533	<0.001
Non-Qataris	237 (53.4%)	8391 (61.2%)	8628	
Total	444 (100%)	13,717 (100%)	14,161	

**Table 3 vaccines-11-01522-t003:** Vaccination status and COVID-19 test results of the study participants.

Vaccination Status	RT-PCR Positive (Cases)	RT-PCR Negative (Controls)	Total
Fully vaccinated	97 (21.8%)	7828 (57.1%)	7925 (55.96%)
Partially vaccinated	1 (0.3%)	10 (0.1%)	11 (0.08%)
Unvaccinated	346 (77.9%)	5879 (42.8%)	6225 (43.96%)
Total	444 (100%)	13,717 (100%)	14,161

## Data Availability

The researchers accessed data through a restricted-access agreement that prevents their sharing with a third party or publicly.

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
