# Peer review of "Pfizer-BioNTech mRNA Vaccine Protection among Children and Adolescents Aged 12–17 Years against COVID-19 Infection in Qatar"

_vaccines, 2023, doi:10.3390/vaccines11101522_

Round 1

Reviewer 1 Report

Thank you for this concise, straightforward study on the efficacy of COVID-19 immunizations (mRNA) in adolescents in Qatar. Overall, I believe that the study is well conducted and has more strengths than weaknesses. The universal availability of vaccine and testing during the study time period provides a best-case scenario for vaccine efficacy. Here are some comments to help further shape your manuscript.

Introduction - It is very inclusive and could probably be summarized a little bit better. It is two pages long and contains some common knowledge facts regarding COVID-19 and immunizations. 

As I am not familiar with the timeline of the pandemic in Qatar, I would like you to provide a little more background information into the availability of over-the-counter testing during this time period in your country. If nearly all testing in your country is via PCR, then this should be highlighted as a strength. Since your study design relies on people being tested, if a home test did not result in the same contract tracing as the PCR, you may be underestimating cases. On the other hand, if only PCR testing was occurring, your results are very strong. 

Additionally, I think you should comment on how non-pharmacologic interventions may have impacted your study population. You address that they were variable in lines 171-175 but do you know if Qatar was more restrictive or opened up more quickly than other countries. This might be addressed in your discussion where you compare your results to studies in other populations. 

In line 204, you discuss how the population is divided into three categories but you actually list 4. Table 1 only has three categories I believe you need to reconcile this section of the manuscript as it is a bit confusing.

You show a very large increase in the number of cases in vaccinated individuals after 90 days. Can you further divide this group, 90 days to 120, 120 to 150? Or can you comment on the latest that adolescents were followed after the completion of the vaccine series?

In your discussion, you compare your study and its vaccine efficacy to other studies that have been published in this area. I think that you need to state which immunizations were delivered in each of these studies to allow the reader to know that you are comparing like groups. Furthermore, in the study from Singapore you refer to the immunization as Comirnaty, the only time you use the trade name. I suggest that you switch this to the generic designation as you have used throughout the paper. 

For your statement of the VISION network study, please state which country that was performed in. 

For your strengths, I would suggest that you highlight the universal access to the immunization and testing in your country. Additionally, I would highlight that your study also looked at testing people who were contacts and thus likely found many asymptomatic patients. These facts would suggest that you are testing vaccine efficacy versus infection and not symptomatic infection, hospitalization or those seeking healthcare.

Throughout the paper, the manner in which you write COVID-19 is different. Please standardize all use of the term to COVID-19.

Author Response

  1. The timeline of the pandemic in Qatar has been included in the Introduction lines 42-53; 108-112
  2. Nearly all testing in the country is done using Rt-PCR, this has been highlighted in lines 58-59 and in 366-367 as a strength.

  3. Since our study design relies on people being tested using RT-PCR, a home test would not result in the same contract tracing as with RT-PCR, which may lead to underestimating cases. However, since nearly all testing in the country is done using Rt-PCR, won't be huge numbers. lines 372-377
  4. We have commented on how non-pharmacologic interventions would not have impacted our study population in lines 181-185

  5. In line 216, we discussed how the population is divided into three categories but listed 4. This has been corrected.

  6. A very large increase in the number of cases in vaccinated individuals after 90 days has been shown. Can you further divide this group, 90 days to 120, 120 to 150? We had opted for this classification of duration to maintain comparability with other studies.

  7. In your discussion, you compare your study and its vaccine efficacy to other studies that have been published in this area. I think that you need to state which immunizations were delivered in each of these studies to allow the reader to know that you are comparing like groups. Included 

  8. in the study from Singapore you refer to the immunization as Comirnaty, the only time you use the trade name. I suggest that you switch this to the generic designation as you have used throughout the paper. --corrected

  9. the VISION network study, please state which country that was performed in. -- added

  10. For your strengths, I would suggest that you highlight the universal access to the immunization and testing in your country. Additionally, I would highlight that your study also looked at testing people who were contacts and thus likely found many asymptomatic patients. These facts would suggest that you are testing vaccine efficacy versus infection and not symptomatic infection, hospitalization or those seeking healthcare.- added

  11. standardized use of the term COVID-19 throughout the paper. 

Reviewer 2 Report

Dear authors,

I have completed my review of your manuscript titled "Pfizer-BioNTech mRNA vaccine protection among children and adolescents aged 12–17 years against COVID-19 infection in Qatar."

Your study evaluates the 30µg formulation of the Pfizer-BioNTech (BNT162b2) COVID-19 vaccine and its effectiveness against pre-Omicron variants among children and adolescents aged 12-17 years in Qatar. The choice of a test-negative matched case-control study with a sample size of 14,161 children/adolescents, and the conclusion of a 79% effectiveness rate against SARS-CoV-2 is particularly noteworthy.

The manuscript is well-constructed and offers significant insights. The examination of the Pfizer-BioNTech COVID-19 vaccine's effectiveness in a specific age group against targeted virus variants is valuable, especially considering the global emphasis on vaccine coverage in the fight against the pandemic.

I offer the following suggestions to enhance the quality of the manuscript:

1. The focus on pre-Omicron variants is clear. However, with the ever-evolving nature of the virus, it would be enlightening to gauge the vaccine's efficacy against subsequent strains. Could you discuss potential avenues to extend your analysis to the post-Omicron era?

2. The chosen age group (12-17 years) is pertinent. Yet, assessing the vaccine's effectiveness in younger children or dissecting its efficiency within age sub-groups of your current range could enrich the study. Please indicate whether data on younger age groups is accessible within your dataset.

3. I observed that adverse events or potential side effects associated with the vaccine receive minimal attention. While your main objective is to ascertain vaccine effectiveness, comprehending possible risks is integral for a holistic view and decision-making. The strong endorsement for vaccination in your conclusion is well-justified by your findings. Still, it might be enhanced with a detailed discussion on safety in tandem with efficacy. Could you delve deeper into this aspect?

4. Your study spans June to November 2021. While this period is significant, further extension or subsequent follow-ups could shed light on enduring efficacy and the prospective requirement for booster shots. If extending the timeframe isn't feasible, perhaps elucidate how the study's current temporal scope can serve both contemporary and future readers.

Thank you for your contribution to the field, and I look forward to your revisions.

Author Response

1. The focus on pre-Omicron variants is clear. However, with the ever-evolving nature of the virus, it would be enlightening to gauge the vaccine's efficacy against subsequent strains. Could you discuss potential avenues to extend your analysis to the post-Omicron era?

 We have the potential to extend our analysis to post Omicron era as well however that was not our objective hence, beyond the scope of this manuscript.

2. The chosen age group (12-17 years) is pertinent. Yet, assessing the vaccine's effectiveness in younger children or dissecting its efficiency within age sub-groups of your current range could enrich the study. Please indicate whether data on younger age groups is accessible within your dataset.

In the current data set only children 12-17 years were extracted, however the national data base consists of even smaller children hence we can do a similar study on children aged 5 years and above.

 3. I observed that adverse events or potential side effects associated with the vaccine receive minimal attention. While your main objective is to ascertain vaccine effectiveness, comprehending possible risks is integral for a holistic view and decision-making. The strong endorsement for vaccination in your conclusion is well-justified by your findings. Still, it might be enhanced with a detailed discussion on safety in tandem with efficacy. Could you delve deeper into this aspect?

Added major highlights on AEFI with COVID vaccines in the study population during that period.

 4. Your study spans June to November 2021. While this period is significant, further extension or subsequent follow-ups could shed light on enduring efficacy and the prospective requirement for booster shots. If extending the timeframe isn't feasible, perhaps elucidate how the study's current temporal scope can serve both contemporary and future readers.

Currently extending the time frame is not feasible as this was used to obtain the IRB approval, however further extension or subsequent follow-ups could be taken up as per your suggestion.

Reviewer 3 Report

This is a population-based case-control study to measure the protective efficacy of Pfizer/BioNTech mRNA vaccine against COVID-19 in Qatari children and adolescents aged 12-17 years. The study was conducted between 1 June and 30 November 2021 and, therefore, it by definition does not measure vaccine efficacy against Omicron (maybe such a study will follow?) but against the prevailing pre-omicron variants. Still, while a bit outdated, it is a worthwhile communication.

The study is carefully conducted and the report is well written. However, some questions remain.

The Abstract says that the specimens (from 14,161 children adolescents, which is a good number) were collected as part of contact tracing. On the other hand, the Methods state that children who were tested as part of screening were excluded. I find it difficult to understand the difference between contact tracing and screening. This should be clarified.

Furthermore, the methods state that those seeking health care were excluded. Why?

The key result was a finding of vaccine effectiveness of 79%, but it is unclear against what. Is this against asymptomatic infection or were some of the subjects that tested positive ill? Why were they then detected in contact tracing rather than health care seeking? Were any of the subjects hospitalized?

This otherwise good paper needs some input from a clinician.

Author Response

  1. The Abstract says that the specimens (from 14,161 children adolescents, which is a good number) were collected as part of contact tracing.- statement has been corrected
  2. The key result was a finding of vaccine effectiveness of 79%, but it is unclear against what. Is this against asymptomatic infection or were some of the subjects that tested positive ill? against SARS CoV2 infection
  3. Why were they then detected in contact tracing rather than health care seeking? the data was extracted from the National COVID 19 case investigation and contact tracing database.
  4. Were any of the subjects hospitalized? No